

# The importance of sponges and mangroves in supporting fish communities on degraded coral reefs in Caribbean Panama

Janina Seemann[1], Alexandra Yingst[1,2], Rick D. Stuart-Smith[3], Graham J. Edgar[3] and Andrew H. Altieri[1,4]

[1] MarineGEO, Smithsonian Tropical Research Institute, Panamá, Republic of Panama
[2] University of Pittsburgh, Pittsburgh, PA, United States of America
[3] Institute for Marine and Antarctic Studies, University of Tasmania, Hobart, Tasmania, Australia
[4] Department of Environmental Engineering Sciences, Engineering School of Sustainable Infrastructure and Environment, University of Florida, United States of America

## ABSTRACT

Fish communities associated with coral reefs worldwide are threatened by habitat degradation and overexploitation. We assessed coral reefs, mangrove fringes, and seagrass meadows on the Caribbean coast of Panama to explore the influences of their proximity to one another, habitat cover, and environmental characteristics in sustaining biomass, species richness and trophic structure of fish communities in a degraded tropical ecosystem. We found 94% of all fish across all habitat types were of small body size ($\leq$10 cm), with communities dominated by fishes that usually live in habitats of low complexity, such as Pomacentridae (damselfishes) and Gobiidae (gobies). Total fish biomass was very low, with the trend of small fishes from low trophic levels over-represented, and top predators under-represented, relative to coral reefs elsewhere in the Caribbean. For example, herbivorous fishes comprised 27% of total fish biomass in Panama relative to 10% in the wider Caribbean, and the small parrotfish *Scarus iseri* comprised 72% of the parrotfish biomass. We found evidence that non-coral biogenic habitats support reef-associated fish communities. In particular, the abundance of sponges on a given reef and proximity of mangroves were found to be important positive correlates of reef fish species richness, biomass, abundance and trophic structure. Our study indicates that a diverse fish community can persist on degraded coral reefs, and that the availability and arrangement within the seascape of other habitat-forming organisms, including sponges and mangroves, is critical to the maintenance of functional processes in such ecosystems.

# INTRODUCTION

Coral reef fishes are useful model communities for exploring drivers of species diversity at landscape and regional scales (*Galzin et al., 1994*; *Fabricius et al., 2005*; *Knowlton et al., 2010*; *Wilson et al., 2010*). They are sensitive to changes in habitat and anthropogenic

Corresponding author
Janina Seemann,
seemannja@gmail.com,
seemannj@si.edu

impacts—a particular concern given that fishes play an important role in coral reef ecosystems, and declines of coral reef fishes threaten people's livelihoods and food security (*Cesar, 2000*; *Cesar, Burke & Pet-Soede, 2003*; *Bellwood et al., 2004*; *Paddack et al., 2009*). A variety of human impacts are responsible for coastal degradation, including habitat destruction, eutrophication, and sedimentation (*Hughes, 1994*; *Jackson et al., 2001*; *Aronson et al., 2003*). Climate change has contributed to ecosystem decline through coral die-off from bleaching, hypoxia events and storms (*Wilson, 2006*; *Alvarez-Filip et al., 2009*; *Wilson et al., 2010*; *Altieri et al., 2017*). The consequences of these events are structural collapses and habitat homogenization in coral reefs, effects which have a variety of potential direct and indirect implications for resident organisms (*Bell & Galzin, 1984*; *Jackson et al., 2001*; *Kuffner et al., 2007*; *Wilson et al., 2010*; *Alevizon & Porter, 2015*; *Mora, 2015*).

Additional factors contributing to declining reef fish abundances are unsustainable fisheries and increasing demand for fish products for a growing human population (*Hodgson, 1999*; *Jackson et al., 2001*; *Zaneveld et al., 2016*). The overexploitation and disproportionate targeting of large size classes and high trophic levels affect fish population structure, growth, and reproduction, and contribute to a trophic imbalance and shifts in trait composition in the reef fish community (*Hixon, Johnson & Sogard, 2014*). This in turn has led to further changes in habitat structure, phase shifts from coral to algal communities, and decreasing ecosystem stability (*Saila, Kocic & McManus, 1993*; *Jennings & Lock, 1996*; *White & Jentsch, 2001*).

Reef fish populations have also been negatively affected by the loss of coastal habitats that provide important nurseries (*Nagelkerken et al., 2000*). The nursery-role concept suggests that many reef fishes (e.g., families Lutjanidae, snappers; Serranidae, groupers; Haemulidae, grunts) have life cycles that include seagrass meadows and mangroves as nursery and feeding grounds (*Beck et al., 2001*; *Nagelkerken et al., 2002*; *Unsworth et al., 2008*; *Ley, 2014*; *Serafy et al., 2015*). Seagrass meadows and mangrove forests have high primary and secondary productivity relative to unvegetated substrates, and support high diversity and abundances of reef fishes (*Nagelkerken et al., 2000*; *Beck et al., 2001*; *Mumby et al., 2004*). Many fish species on coral reefs therefore depend on the connectivity to, and integrity of, associated habitats.

Our study region, the Caribbean Sea, has experienced declining reef fish populations as a result of pollution, ecosystem degradation and unsustainable reef fisheries (*Hughes, 1994*; *Gardner et al., 2003*; *Bellwood et al., 2004*; *Paddack et al., 2009*). These problems appear particularly prominent at our focal study area in Bocas del Toro on the Caribbean coast of Panama, where rapid human population growth connected with agriculture (banana industry) and tourism has accelerated the decline of water quality and the physical destruction of reefs, and has increased fishing pressure (*Guzmán & Jiménez, 1992*; *Collin, 2005*; *D'Croz, Rosario & Gondola, 2005*; *Cramer, 2013*; *Aronson et al., 2014*; *Seemann et al., 2014*). Bocas del Toro encompasses a coastal coral reef-seagrass-mangrove system in a semi-enclosed lagoon. It is composed of six major islands and the mainland that surround the Almirante Bay, and includes mangroves fringing the mainland and mangrove islands scattered across the bay (*Collin, 2005*; *Guzmán et al., 2005*). Reefs are typically dominated by corals with a high stress tolerance, including *Porites furcata* in shallow

(1–4 m) and *Agaricia* spp. (>3 m) in the deeper areas (*Seemann, 2013*; *Aronson et al., 2014*; *Seemann et al., 2014*). Associated seagrass meadows are dominated by *Thalassia testudinum* (turtlegrass). Mangrove fringes are comprised of *Rhizophora mangle* (red mangrove). Several rivers, creeks and oceanic inlets discharge sediments and nutrients into the bay (*Beulig, 1999*; *Collin, 2005*). Bleaching and low oxygen events occur regularly due to lagoonal characteristics including retention of warm water and depletion of oxygen (*Kaufmann & Thompson, 2005*; *Seemann et al., 2014*; *Altieri et al., 2017*). Bocas del Toro reefs potentially represent a model system for improving predictions relevant throughout the region due to their exposure to common stressors, such as high terrigenous runoff, nutrient levels, and overfishing, that are afflicting other coral reefs in the Caribbean (*Riegl et al., 2009*; *Sammarco & Strychar, 2009*; *Leinfelder et al., 2012*; *Aronson et al., 2014*).

This study aims to characterize the ecosystem attributes that facilitate the maintenance of essential functions, biodiversity and biomass of coral reef fish communities in a degraded ecosystem. Specifically, we (1) quantify the fish community at 67 sites in five bioregions of the Caribbean to assess the status of our focal study system in Bocas del Toro along a gradient of ecosystem degradation and over-fishing, (2) examine the effects of proximity of mangroves and seagrass for fish communities on coral reefs, and (3) identify characteristics of coral reef habitat that are positively related to biomass, abundance and structure of the fish community. Addressing these objectives contributes to a better understanding of how landscape-scale features underlie the resilience of degraded coastal habitats.

## METHODS

### Study system

To place results within the wider regional context, we conducted fish surveys at reefs with different fishery management restrictions in different Caribbean ecoregions.

In our focal study areas of Bocas del Toro on the Caribbean coast of Panama (Fig. 1), we also conducted comprehensive surveys of fish communities, benthic surveys, and water quality measurements in adjacent seagrass and mangrove fringe areas. All data from Bocas del Toro were collected from May to July 2015. Data for the other Caribbean regions were collected from 2012 to 2015. Research was conducted under a Scientific Permit from the Ministry of the Environment Panama (MiAmbiente) and Autoridad de los Recursos Acuáticos de Panamá (ARAP) with the Number: SE/APO-1-15 & 10b.

### Caribbean data set
#### Fish surveys

We conducted visual fish surveys using the Reef Life Survey (RLS) method 1 protocol (*Edgar & Stuart-Smith, 2014*) at 67 reef sites in the following five ecoregions (*Spalding et al., 2007*): Southern Caribbean (14 sites, Bonaire), Southwestern Caribbean (31 sites, Bocas del Toro, Kuna Yala, Archipelago of San Andres), Greater Antilles (one site, Grand Cayman), Floridian (17 sites, Florida Keys) and Bahamian (four sites, Turks and Caicos Islands). Surveys involved underwater visual censuses by scuba divers at reef sites (each with 2–6 replicate transects) in depths of 1 to 35 m. Divers counted and assigned all fish species observed within binned size-classes along a $50 \times 5$ m belt transect ($250$ m$^2$). All

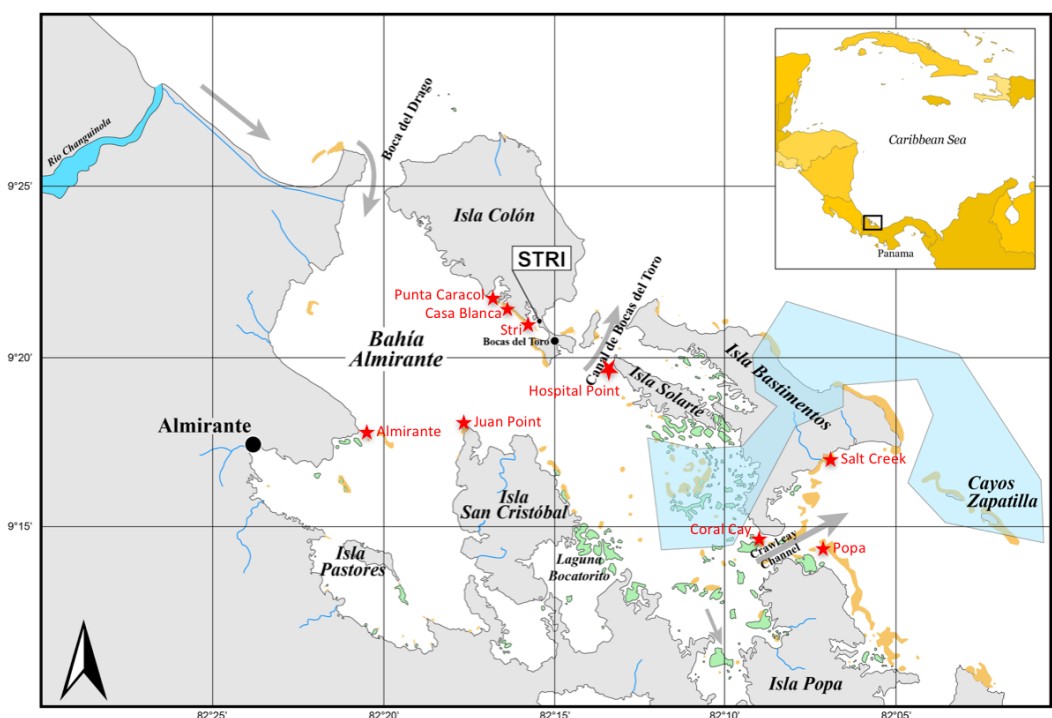

**Figure 1 Sampling sites in Bocas del Toro.** Three reef sites (Punta Caracol, Casa Blanca, Almirante) possess close connectivity with mangrove habitat (within 100 m), three sites (STRI, Juan Point, Coral Cay) represent reef sites further away from mangroves (100–250 m), and three reef sites (Popa, Salt Creek, Hospital Point) are not closely connected to mangroves (>750 m). Yellow areas are reefs and green areas are mangrove islands, gray is land, white is ocean, blue is river and blue polygon is a poorly enforced MPA.

fishes sighted on each transect were recorded on a waterproof datasheet as the diver swam along the transect at approximately 2 m min$^{-1}$. We identified fish species to the highest taxonomic resolution possible, and estimated body length for each individual. The order of priority for recording accurately was to first ensure all species observed along transects were included, then individuals of larger or rare species were accurately counted, then estimates were made of abundances of common species. If an individual could not be identified underwater, a photograph was taken for later identification. Abundance, size and species identity were used to estimate biomass in kg ha$^{-1}$ using conversion factors provided by Fishbase (http://www.fishbase.com), as described by *Edgar & Stuart-Smith (2014)*.

## Bocas del Toro data set
### Fish surveys

The RLS method employed on coral reefs in five ecoregions was also applied in seagrass and mangrove habitats located within 250 m of reef sites in Bocas del Toro. Seagrass sites ranged in depth from 1 m to 4 m, whereas mangrove fringe root systems had maximal depth of 2 m. Mangrove surveys were conducted amongst the mangroves prop roots below the upper intertidal fringe with counts and size estimates made for all fishes in a 5 m wide belt within the mangrove root system. Two 50 m transects were laid end-to-end along
the mangrove fringe given that side-by-side replicate transects typical of the RLS protocol could not be applied within mangrove root habitats.

### Habitat assessment

We conducted benthic surveys to characterize coral reef and seagrass bed habitats. Reef and seagrass benthos were analyzed with 20 photo quadrats (0.5 m$^2$), which were taken every 2.5 m along the 50 m long transects at each site. Photos were analyzed via point counting using the Coralnet annotation tool (coralnet.ucsd.edu). A total of 25 points were randomly distributed on each photo and categorized. Substratum categories for analyses were: healthy hard coral, bleached hard coral, recently-dead coral, anemones, non-calcifying corals (including hexacorals and octocorals), sponges, sessile worms (tube worms, mostly polychaetes), zoanthids, rubble, sand, rock, calcifying algae, seagrass and macroalgae. If sessile organisms were too small for identification or obscured by dark shadows, then they were excluded from the dataset. In addition, the distance between reef sites surveyed and nearest mangrove was measured using GPS coordinates (Table 1).

### Water quality measurements

Water quality was assessed by quantifying temperature (°C), salinity (psu), water depth (m), total dissolved solids (TDS, mg L$^{-1}$), dissolved oxygen (mg L$^{-1}$), pH, turbidity (FNU), chlorophyll (µg L$^{-1}$), blue–green algae concentrations (µg L$^{-1}$), and dissolved organic matter (fDOM, RFU) with an Exo2 multiparameter sonde (Xylem brand; YSI, Yellow Springs, OH, USA). The sonde was positioned ∼10 cm above the bottom in each habitat (reef, seagrass and mangrove fringe). Measurements were recorded at intervals of 1–6 min over a time period of at least 30 min during the fish surveys, and constrained to the mid-day hours 10 am–3 pm, hence measurements were subject to the daily variability of weather conditions or tidal cycles.

## Data analyses

The Caribbean reef fish data set was used to characterize the fish community in relation to the protection status of the sites. All 67 sites from the five different ecoregions were individually classed by management type using the criteria of *Edgar & Stuart-Smith (2014)*: NTZ (no take zones, $n = 27$), RZ (restricted zones that allow some methods of fishing within a MPA, $n = 19$) and OZ (open zones where fishing is unrestricted, $n = 12$). These data were compared to data from Bocas del Toro (OZ, $n = 9$). Replicated surveys from each site were averaged.

Data from the fish surveys were used to calculate fish community metrics, including total abundance (density), abundance of major fish families within size bins ($\leq 10$ cm; $>10$–$20$ cm; $>20$ cm), total biomass, biomass of fishes $\leq 10$ cm, and total species richness. We also calculated the mean trophic level as an abundance weighted mean of the reef fish community by multiplying the trophic level of each species by their log abundance, summing these values across species recorded on a transect, and dividing by the total log abundance of all fishes on the transect. The classification of the trophic level (2–5) for each species was based on feeding strategy: herbivores and detritivores (2–2.1), omnivores (2.2–2.7), low-level carnivores (2.8–3.4), mid-level carnivores (3.5–3.9) and high-level carnivores

Seemann et al. (2018), *PeerJ*, DOI 10.7717/peerj.4455

**Table 1  Major habitat characteristics and location of monitoring sites.** Sites 7, 8 and 9 did not have mangroves in close proximity (≤250 m); site 9 also did not have a seagrass meadow close to the reef, effective juvenile habitats were calculated from the fish abundance in mangroves (w/o small bodied fish) compared to reef fish abundance.

| | Site | Coordinates Lat | Coordinates Long | Depth reef (m) | Depth seagrass (m) | Distance Reef-mangrove (m) | Sponge cover % | Live hard coral cover % | Hard substrate % | Reef fish Biomass kg ha$^{-1}$ | Seagrass fish Biomass kg ha$^{-1}$ | Mangrove fish Biomass kg ha$^{-1}$ | Seagrass as effective juvenile habitats% | Mangroves as effective juvenile habitats% | Reef fish abundance ha$^{-1}$ | Seagrass fish abundance ha$^{-1}$ | Mangrove fish abundance ha$^{-1}$ | Reef fish richness | Seagrass fish richness | Mangrove fish Richness |
|---|---|---|---|---|---|---|---|---|---|---|---|---|---|---|---|---|---|---|---|---|
| 1 | Punta Caracol | 9.3757° | −82.2997° | 3 | 2 | 65 | 9.5 | 41.5 | 33 | 201 | 25 | 111 | 109 | 131 | 12,929 | 2,820 | 17,423 | 38 | 12 | 21 |
| 2 | Casa Blanca | 9.3588° | −82.2737° | 3 | 1 | 70 | 17.5 | 2.5 | 71 | 67 | 32 | 47 | 102 | 73 | 18,741 | 67,660 | 17,570 | 30 | 9 | 16 |
| 3 | Almirante | 9.2900° | −82.3429° | 3 | 2 | 90 | 19.5 | 36.5 | 71 | 206 | 2 | 202 | 13 | 59 | 11,105 | 2,560 | 1,202,510 | 28 | 6 | 15 |
| 4 | STRI Point | 9.3483° | −82.2625° | 3 | 4 | 120 | 19.5 | 3.0 | 57 | 257 | 24 | 15 | 49 | 29 | 71,076 | 73,153 | 42,390 | 35 | 15 | 19 |
| 5 | Juan Point | 9.3003° | −82.2921° | 4 | 1 | 170 | 17.6 | 46.4 | 69 | 94 | 14 | 32 | 86 | 29 | 24,045 | 31,760 | 200,660 | 30 | 11 | 10 |
| 6 | Coral Cay | 9.2435° | −82.1478° | 5 | 2 | 230 | 2.0 | 16.0 | 51 | 25 | 12 | 11 | 54 | 139 | 1,717 | 50,850 | 42,060 | 25 | 7 | 9 |
| 7 | Popa | 9.2336° | −82.1120° | 3 | 1 | 700 | 1.1 | 26.9 | 61 | 60 | 2 | | 23 | | 2,608 | 560 | 17,423 | 24 | 9 | |
| 8 | Salt Creek | 9.2815° | −82.1012° | 6 | 2 | 950 | 0.5 | 24.8 | 99 | 13 | 0 | | 105 | | 1,688 | 1,290 | | 15 | 12 | |
| 9 | Hospital Point | 9.3326° | −82.2220° | 5.5 | | 900 | 0.5 | 96.0 | 33 | 12 | | | | | 1,946 | | | 16 | | |

(4–4.5) (classification and values obtained from Fishbase; http://www.fishbase.org). We also compared preferred substrate types and resilience factors of the fish species (values obtained from Fishbase), the latter estimated from population doubling time (low, medium, high). Fish community metrics were averaged within sites and compared among regions for significant differences using one-way ANOVA or a Student's $t$-test.

For the Bocas del Toro dataset only, we assessed whether mangroves and seagrasses provided juvenile or alternative habitats to coral reefs by comparing the abundance (log transformed) and composition of fishes in the different habitat types. We assumed that higher abundances of fishes amongst mangroves and seagrasses compared to reefs, and high species similarity, indicates higher likelihood and magnitude of migration and exchange rates. We excluded small-bodied species (maximum total length ≤12.5 cm), which are presumably non-migratory fish species (*Dahlgren et al., 2006*), such as *Apogon townsendi* (belted cardinalfish), *Coryphopterus* spp., and *Canthigaster rostrata* (caribbean sharpnose-puffer). A principal component analysis (PCA) on correlations (fish abundance log transformed, only fish >12.5 cm) was used to compare differences in the fish communities between reefs at different distances to mangroves and seagrass.

We also tested for correlations between environmental factors and the reef fish community metrics across all sites. Environmental factors included reef cover, cover of the seagrass benthos, distance to mangroves and water quality parameters. Fish metrics included species richness, biomass, size structure, the abundances of individual fish species, tropic levels, preferred substrate types and resilience factors. Data were characterized using a scatterplot matrix (see Supplemental Information 4) and nonparametric Spearman's tests for pairwise correlation probabilities. For all statistical analyses, fish abundance data were log-transformed to down-weight the extremely high abundance of a few fish species (*Edgar et al., 2014*). All statistical analyses were conducted using JMP Software 13.01.

## RESULTS

### Characteristics of the fish community

We recorded a total of 77 fish species across all habitats in Bocas del Toro, of which 61 species were found on coral reefs. The average mean richness per transect was $29 \pm 7$ (SD) species. This value was low compared to our other Caribbean survey sites which had a mean richness per transect of $52 \pm 4$ species (with a cumulative total of 196 species recorded in the whole Caribbean) (*Stuart-Smith et al., 2013*; *Edgar & Stuart-Smith, 2014*). Fish biomass on Bocas del Toro reefs ($71 \pm 63$ kg ha$^{-1}$) was also lower than on other Caribbean reefs in both no-take zones and MPAs with restricted fishing (ANOVA, $P = 0.02$ and $0.001$, respectively), although the difference was not significantly lower for open zones ($P > 0.05$) (Fig. 2A). Moreover, the range of total observed fish biomass in Bocas del Toro (30–1,350 kg ha$^{-1}$) represents the lowest range found amongst fish surveys conducted in the Caribbean, which were 140–5,930 kg ha$^{-1}$ elsewhere.

The biomass of herbivorous, omnivorous and detrivorous fishes in Bocas del Toro (trophic level 2–2.7) was 37% of the total biomass and 76% of all individual fishes counted (Fig. 2B). Herbivores alone comprised $27 \pm 3.5\%$ (SD) of biomass versus $10 \pm 4\%$ across

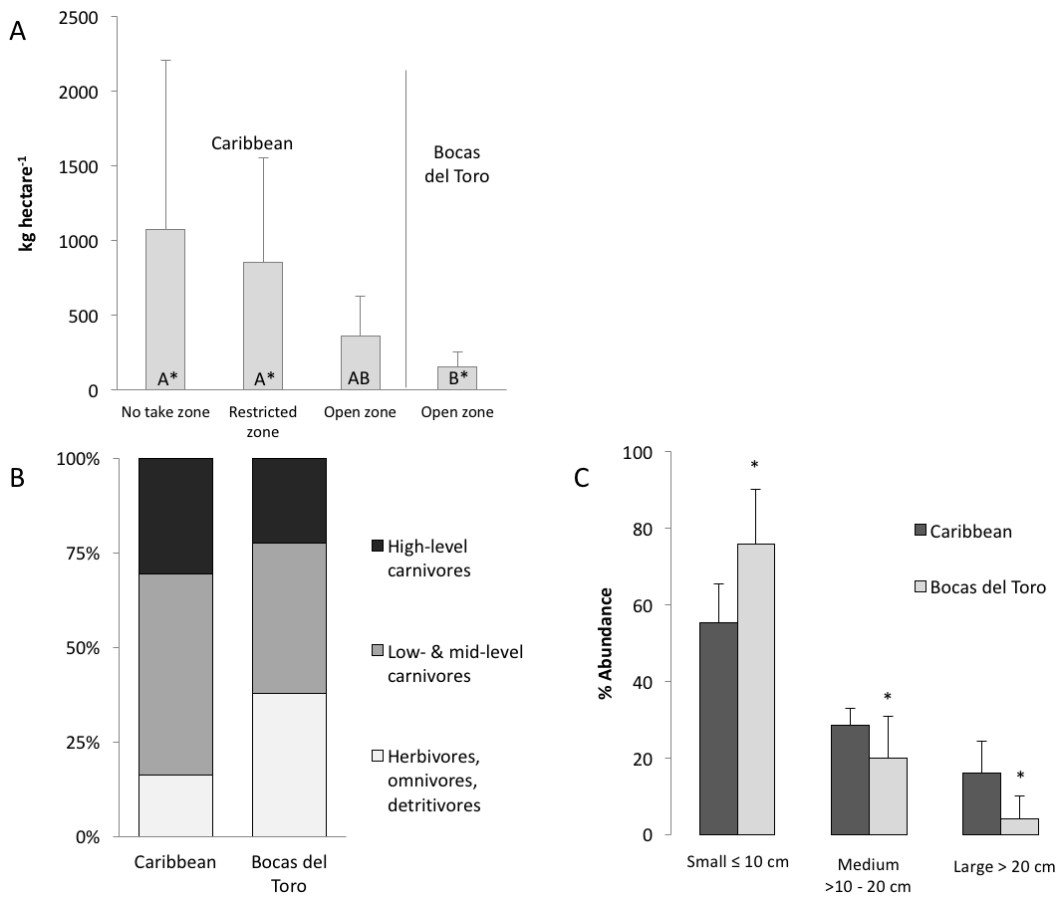

**Figure 2 Biomass and composition of the fish community in the Caribbean and Bocas del Toro.** (A) The comparison of the total biomass (AVR ± SD) from RLS conducted across the Caribbean, divided in no take zones, restricted zone and open zones, and open zones in Bocas del Toro, groups with different letters are significantly different. (B) Distribution of trophic guilds based on total biomass: high-level carnivores (trophic level 4–4.5), low and mid-level carnivore (trophic level 2.8–3.9), herbivores, omnivores and detrivores (trophic level 2–2.7). (C) The abundance of fish subdivided in size classes (AVR ± SD), which are indicative of fishing pressure (a skew towards smaller body size implies fishing). Asterisk represents significant differences between size abundance data from Bocas and the Caribbean.

the wider Caribbean. Pomacentridae (damselfishes) and Scarinae (parrotfishes) were the predominant taxa in terms of biomass. *Scarus iseri* (striped parrotfish) contributed 72% of the parrotfish biomass. High-level carnivores contributed 22 ± 3.5% of total fish biomass, versus 31 ± 4% elsewhere in the Caribbean. Dominant high-level carnivores in Bocas del Toro were *Caranx ruber* (bar jack), *Cephalopholis cruentata* (graysby), *Hyplopectrus nigricans* (black hamlet) and *Scomberomorus regalis* (cero). There was a trend for fish communities in Bocas del Toro to exhibit a greater proportion of total biomass comprised of herbivores, omnivores and detrivores (trophic level: 2–2.7), and a lower proportion comprised of high-level carnivores (trophic level: 4–4.5), relative to other Caribbean reefs, although the difference was not significantly different for either group (Fig. 2B).

A total of 94% of all fishes observed across all habitat types (reef, seagrass, mangrove) in Bocas del Toro were in the smallest size class (≤10 cm length). Fishes ≤10 cm represented 59% of the total biomass within the reefs. The abundance of fishes within the smallest size class (≤10 cm) was significantly higher in Bocas del Toro than other Caribbean reefs (ANOVA, $P < 0.0001$), whereas the abundances of medium- (>10–20 cm) and large- (≥20 cm) sized fishes were significantly lower (ANOVA, $P < 0.0001$) (Fig. 2C). This pattern was also evident when comparing Bocas del Toro with other Caribbean reefs across fish families (Table 2).

## Relationships between environmental factors and fish community composition

Some environmental parameter and habitat factors were associated with the reef fish community metrics in Bocas del Toro. Some environmental factors were not independent, as sponge cover was negatively correlated with the distance to mangroves and also positively to chl $a$ ($R^2 = 0.60$ and $R^2 = 0.70$, respectively, $P < 0.01$). The other water quality parameters were not found to correlate with any fish community or species metrics. Sponge cover was the strongest positive correlate among all environmental parameters for species richness ($R^2 = 0.5$, $P < 0.01$), small fish ≤10 cm biomass ($R^2 = 0.85$, $P < 0.01$), and trophic level of the fish community ($R^2 = 0.89$, $P < 0.01$). The abundance of *Abudefduf saxatilis* (sergeant major) was significantly correlated with sponge cover ($R^2 = 0.62$, $p = 0.0027$). Survey sites characterized by high sponge cover and low distance to mangroves were characterized by fishes such as *Abudefduf saxatilis* (sergeant major), *Hypoplectrus nigricans* (black hamlet), *Coryphopterus personatus* (masked goby) and *Coryphopterus glaucofraenum* (bridled goby). *Scarus iseri* (striped parrotfish), *Stegastes partitus* (bicolor damselfish) and *Cephalopholis cruentatus* (graysby) had a positive association with recently dead corals, however, the cover of dead corals was negatively correlated with the abundance of most fish species.

Fish species richness on a given reef was positively correlated with richness values in nearby mangroves ($R^2 = 0.76$, $P < 0.05$). The three reef sites with low sponge cover and without mangroves in close proximity (Salt Creek, Popa, Hospital Point) showed lower biomass, abundances and species richness of fishes (Table 1, Figs. 3 and 4). The site without either mangroves or seagrass nearby (Hospital Point) showed the lowest species richness.

Distance to mangroves was also identified as a factor that influenced reef fish communities (Fig. 5), suggesting that proximity to mangroves has a strong influence on reef fish communities, likely by mangroves functioning as effective nursery grounds and as alternative complex habitats. The proportion of carnivorous fishes was higher at the sites closer to the mangroves than those sites that were further away (ANOVA, $P < 0.01$, Fig. 4). However, a more detailed look at the fish communities revealed that the sites at an intermediate distance from mangroves (STRI, Juan Point, Coral Cay) possessed a significantly higher proportion of top-level carnivores than sites that were closer or further away (Fig. 4, ANOVA, $P < 0.01$).

The highest abundances of all fish observed were recorded for the families Pomacentridae (damselfishes) and Gobiidae (gobies) (Table 2). However, Gobiidae were abundant only at

Seemann et al. (2018), *PeerJ*, DOI 10.7717/peerj.4455

**Table 2** Major fish families (only considering >10 counts ha$^{-1}$ in average in one of the size bins).

| | Caribbean | | | | | | Bocas del Toro | | | | | |
|---|---|---|---|---|---|---|---|---|---|---|---|---|
| | Reef | | | Reef | Seagrass | Mangrove | Reef | Seagrass | Mangrove | Reef | Seagrass | Mangrove |
| | ≤10 cm | >10–20 cm | >20 cm | ≤10 cm | ≤10 cm | ≤10 cm | >10–20 cm | >10–20 cm | >10–20 cm | >20 cm | >20 cm | >20 cm |
| Acanthuridae | 317 | 351 | 127 | 113 | 127 | 120 | 233 | 0 | 20 | 0 | 0 | 0 |
| Balistidae | 100 | 380 | 145 | 0 | 0 | 0 | 0 | 0 | 0 | 0 | 0 | 0 |
| Carangidae | 296 | 1,78 | 145 | 321 | 20 | 2,93 | 330 | 80 | 100 | 40 | 0 | 0 |
| Clupeidae | 11,500 | 0 | 0 | 0 | 46,000 | 278,080 | 0 | 0 | 0 | 0 | 0 | 0 |
| Ephippidae | 0 | 80 | 280 | 0 | 0 | 0 | 30 | 0 | 0 | 0 | 0 | 0 |
| Gerreidae | 0 | 30 | 20 | 0 | 600 | 155 | 0 | 0 | 20 | 0 | 0 | 0 |
| Gobiidae | 6,239 | 0 | 0 | 18,182 | 30 | 80 | 0 | 0 | 0 | 0 | 0 | 0 |
| Grammatidae | 434 | 0 | 0 | 0 | 0 | 0 | 0 | 0 | 0 | 0 | 0 | 0 |
| Haemulidae | 1,959 | 1,395 | 160 | 379 | 752 | 823 | 457 | 300 | 70 | 20 | 0 | 0 |
| Holocentridae | 253 | 441 | 50 | 0 | 0 | 0 | 20 | 0 | 0 | 0 | 0 | 0 |
| Inermiidae | 300 | 3,444 | 0 | 0 | 0 | 0 | 0 | 0 | 0 | 0 | 0 | 0 |
| Kyphosidae | 463 | 733 | 160 | 0 | 0 | 0 | 0 | 0 | 0 | 0 | 0 | 0 |
| Labridae | 1,749 | 659 | 96 | 254 | 568 | 580 | 252 | 180 | 80 | 0 | 0 | 0 |
| Loliginidae | 0 | 240 | 0 | 0 | 0 | 0 | 0 | 0 | 0 | 0 | 0 | 0 |
| Lutjanidae | 263 | 800 | 279 | 80 | 137 | 559 | 80 | 20 | 350 | 0 | 0 | 20 |
| Mullidae | 245 | 429 | 229 | 50 | 20 | 0 | 20 | 0 | 0 | 0 | 0 | 0 |
| Pomacentridae | 2,145 | 414 | 20 | 618 | 325 | 123 | 110 | 0 | 0 | 0 | 0 | 0 |
| Scaridae | 741 | 252 | 196 | 494 | 753 | 979 | 333 | 20 | 173 | 80 | 0 | 0 |
| Sciaenidae | 532 | 176 | 40 | 60 | 0 | 0 | 60 | 0 | 0 | 0 | 0 | 0 |
| Serranidae | 855 | 107 | 208 | 297 | 72 | 40 | 93 | 0 | 0 | 0 | 0 | 0 |
| Sphyraenidae | 120 | 2,100 | 180 | 0 | 0 | 40 | 0 | 20 | 40 | 0 | 0 | 100 |
| Tetraodontidae | 247 | 0 | 0 | 193 | 100 | 20 | 0 | 0 | 0 | 0 | 0 | 0 |

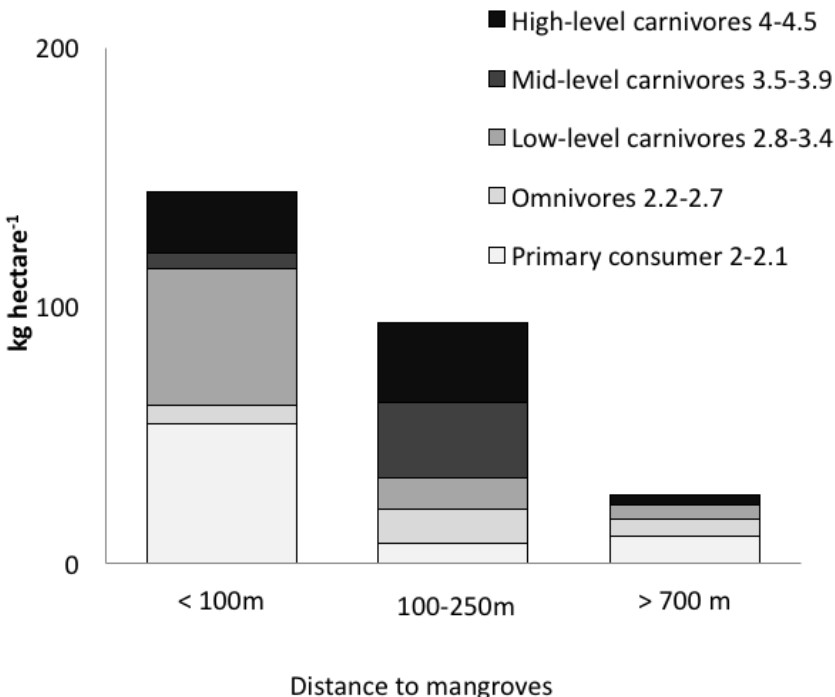

**Figure 3 Biomass of trophic guilds of reef fish.** Data were pooled by sites with a similar distance to mangroves: <100 m (Punta Caracol, Casa Blanca, Almirante), 100–250 m (STRI, Juan Point, Coral Cay) and >700 m (Popa, Salt Creek, Hospital Point) (see Table 1 for detail).

sites close to mangroves. *Coryphopterus personatus* (masked goby) dominated these sites, with abundances up to 13 individuals $m^{-2}$. RLS conducted elsewhere in the Caribbean (e.g., San Andres Archipelago, 350 km distant) revealed much lower densities for the same species (0.2 individuals $m^{-2}$).

Generally, fishes with life cycles closely associated with hard corals (Lewis, 1997), such as Pomacanthidae (angelfishes), were present in very low numbers on the reefs of Bocas del Toro (<1 per transect). Other reef fishes typically associated with hard substrates with a high complexity, such as Balistidae (triggerfishes), Apogonidae (cardinalfishes), Muraenidae (moray eels), Sciaenidae (drums), Pseudochromidae (dottybacks) and Serranidae (grouper), were scarce within the bay (<1 per transect). Many fish species, including those at higher trophic levels, such as *Diodon hystrix* (porcupinefish), *Ginglymostoma cirratum* (nurse shark), *Gymnothorax Funebris* (moray eel), *Lutjanus jocu* (dog snapper), *Ocyurus chrysurus* (yellowtail snapper), *Pomacanthus arcuatus* (gray angelfish) were observed only on reefs with mangroves in close proximity (≤250 m distance).

## DISCUSSION

Our surveys revealed that the fish fauna in Bocas del Toro is depauperate in species richness and biomass by Caribbean standards. We found evidence that the fish community is representative of a degraded and overexploited ecosystem, characterized by numerical dominance of fishes that are small bodied and also typical of habitats of low complexity,

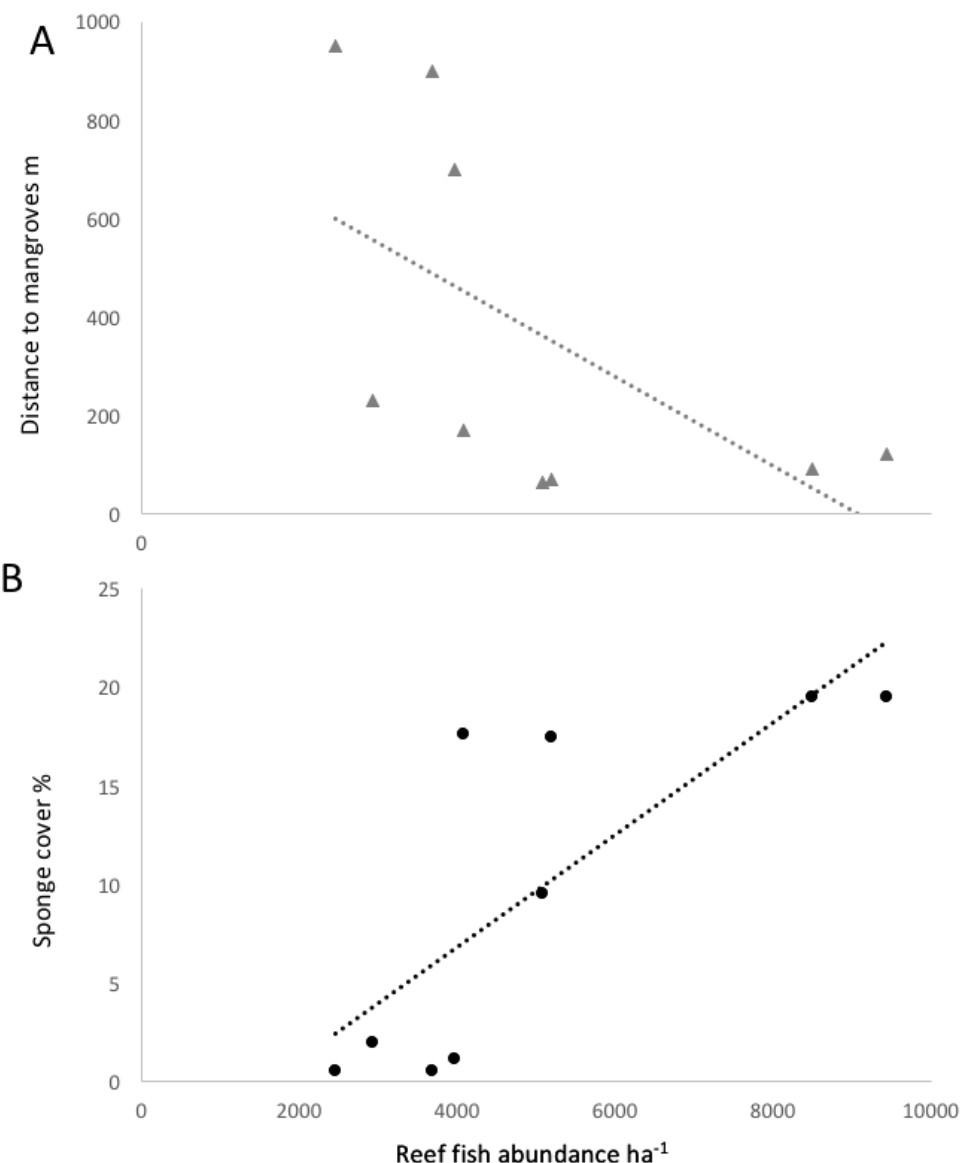

**Figure 4** Correlation of reef fish abundance (w/o small-bodied fish) and (A) distance to mangroves or (B) sponge percent cover.

such as Pomacentridae and Gobiidae, with few representatives of fish families that achieve body sizes targeted by fisheries or that are commonly associated with high-relief coral reefs. Nevertheless, sponge cover and proximity to mangroves were found to be positively correlated with fish species richness, biomass, abundance and trophic level. This pattern suggests that sponges as habitat-forming reef organisms, and mangroves as nursery grounds and alternative habitats, continue to provide critical habitats for the reef fish communities in a degraded ecosystem, and therefore counteract some effects of reef degradation.

Some fishes appeared to be an indicator species for the overall trends observed at our study site. One example is the goby *Coryphopterus personatus,* which forms schools that

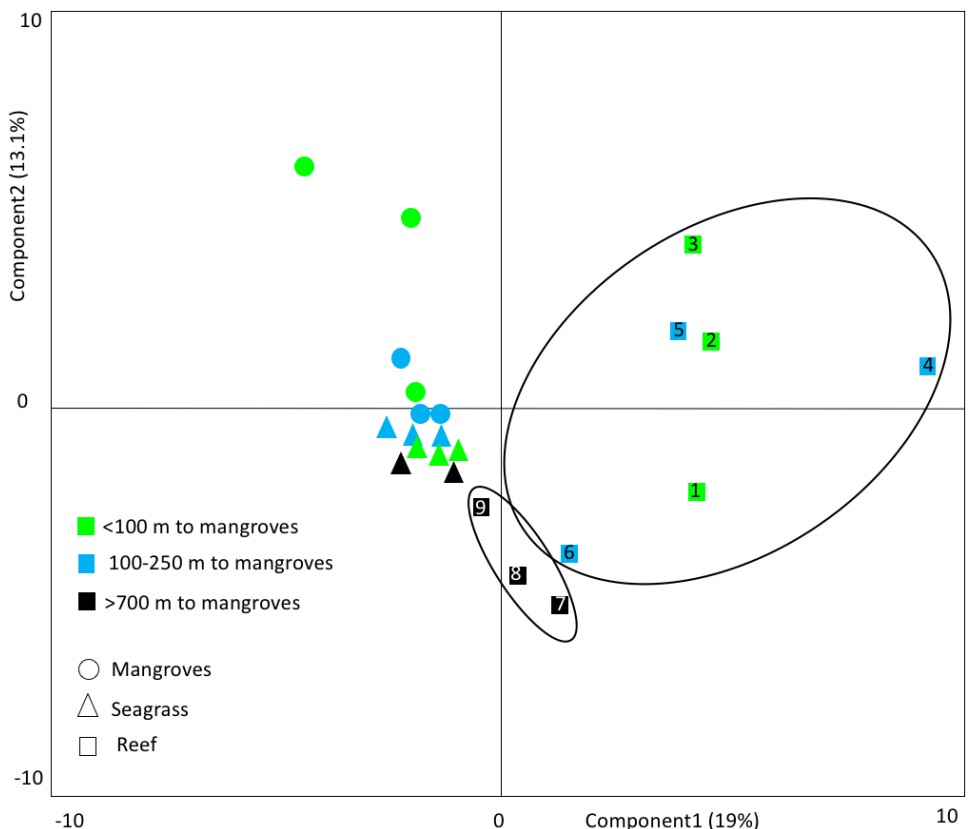

**Figure 5 Principal component analyses.** This PCA shows the clustering of fish communities, considering species composition and abundance (log transformed). Small bodied fish ≤12.5 cm was excluded. Reefs >700 m from mangroves cluster together as a group, separate from the reefs that are closer to mangroves. This indicates that mangrove distance has a strong influence on reef community types. The numbers refer to the site numbers in Table 1.

hover in a vulnerable position above the bottom in extremely high abundances (65-fold higher abundances than in the San Andres Archipelago). Moreover, fish surveys in our Bocas del Toro study area from 2002 revealed densities an order of magnitude lower at 1.2 individuals m$^{-2}$ (*Dominici-Arosemena & Wolff, 2005*). We suggest that this goby is an indicator species for overfished reefs that benefits from a loss of predatory fishes that historically limited their densities. Another indicator species is *Scarus iseri*, which is ecologically important given its role as the predominant herbivorous fish in Bocas del Toro (*Kuempel & Altieri, 2017*). This species likely plays an important role supporting the growth of sponges and corals by cropping competing macroalgae. Third, *Abudefduf saxatilis* was identified as an indicator for sponge cover, which in turn is a factor positively correlated to fish richness, biomass, abundance and relatively high mean community trophic levels.

A degraded reef fish community in Bocas del Toro is evidenced by low total biomass, under-representation of biomass at high trophic levels, and high abundance of small fishes, all classic symptoms of over-fishing (*Pauly et al., 1998*; *Myers & Worm, 2003*). Moreover, the range of total observed fish biomass represents the lowest value found amongst fish

surveys conducted in the Caribbean. High-level carnivores and large fishes are depleted in intense fisheries (*Cinner & McClanahan, 2006*; *Wilson et al., 2010*), causing a skewing of the food web and community size structure. As described by *Wilson et al. (2010)*, the loss of individuals within the largest size classes, which have the highest per capita reproductive output and produce the majority of juveniles, impacts the recruitment of small size classes of juvenile reef fish. Accordingly, we observed that small Haemulidae were rare on Bocas del Toro reefs. Exploitation thus appears to have contributed substantially to the distorted fish community patterns observed at Bocas del Toro (*Guzmán et al., 2005*; *Cramer, 2013*).

Another plausible hypothesis for the low total fish biomass and trophic shifts within the fish community in Bocas del Toro relative to other Caribbean sites is the loss of hard corals (*Turner et al., 1999*; *Wilson et al., 2010*). This in turn results in the loss of shelter and feeding grounds (*Turner et al., 1999*; *Alevizon & Porter, 2015*). This hypothesis was supported by significant negative correlations between the proportions of recently-dead corals and the biomass of fishes, as well as the finding that fish species that are known to associate with hard corals or hard substrate were rare. Moreover, fishes known to live on habitats of low complexity (particularly Pomacentridae and Gobiidae) and grazers (particularly Scaridae and Pomacentridae) occurred in very high abundances (*Booth & Beretta, 1994*; *Bruggemann, Kuyper & Breeman, 1994*).

Herbivores, detritivores and omnivores were overrepresented in the Bocas del Toro fish community compared to elsewhere in the Caribbean. Herbivorous species alone comprised nearly a third of the total fish biomass, which could be explained by a decreased number of predators in the system. Even though most herbivorous fishes were in the smallest size category ($\leq 10$ cm), this group has the potential to control the growth of macroalgae and prevent algal phase shifts, particularly in combination with invertebrate herbivores, such as sea urchins, which are abundant in this system (*Kuempel & Altieri, 2017*). However, if the reduction of live coral cover continues, herbivorous fishes may reach their limits for grazing control (*Williams & Polunin, 2001*; *Williams, Polunin & Hendrick, 2001*). Also, the lack of redundant species within the herbivore functional group is likely to result in low resilience, since a system with a single dominant herbivorous species (e.g., *Scarus iseri* with 72% of biomass in our system) is vulnerable to stressors affecting that species (*Hughes, 1994*; *White & Jentsch, 2001*). The reason for the dominance of one herbivore species may be attributable to the small body size of *S.iseri,* which matures at ~65 mm. It is therefore not a targeted fishery species, and escapes most fishing pressure (*Kuempel & Altieri, 2017*).

Sponges cover up to 20% of substrata, and thus provide considerable physical structure on the Bocas del Toro reefs (*Diaz & Rützler, 2001*; *Loh & Pawlik, 2014*; *Loh et al., 2015*). In the absence of high cover of hard corals, sponges likely play an important role in supporting richness, biomass and expanded trophic levels of the depauperate fish community in our study system. We found evidence for such a positive effect, with an increased abundance of reef fishes with increased sponge cover. Sponges are major determinants of the rugosity and height of the reef (*Diaz & Rützler, 2001*), and thus could be an important driver for fish abundance and species richness in Bocas del Toro as in other Caribbean reef systems (*Gratwicke & Speight, 2005*). Sponges also comprise an important food source

for spongivorous reef fishes, such as some members of Pomacentridae and Scarinae (*Sammarco, Risk & Rose, 1987*; *Dunlap & Pawlik, 1996*; *Pawlik, 1998*; *Souza, Ilarri & Rosa, 2011*). The pomacentrid *A. saxatilis* has been identified to have a functional dependency on sponges through either shelter or other aspects of habitat complexity that sponges provide (*Gratwicke & Speight, 2005*).

Proximity to mangroves was another important positive factor associated with fish communities, as the biomass and species richness of fishes were greater on coral reefs near mangroves. Mangroves are widely recognized for their functions of providing nursery grounds, shelter and food sources for reef fishes (*Laegdsgaard & Johnson, 2001*; *Mumby et al., 2004*). Our study suggests that the positive effect of mangroves as nursery and alternative adult habitats is an important factor maintaining diversity and biomass of the reef fish communities, and that this function remains particularly important in a system as degraded as Bocas del Toro. However, we did not find such evidence for seagrass meadows. Lowest fish species richness, biomass and trophic levels were found on reefs without mangroves in close proximity, presumably because many reef fish species depend on interconnectivity between habitat types (*Ley, 2014*).

Our findings suggest that Bocas del Toro, Panama is a model system for the study of reef fish communities that are associated with high levels of anthropogenic stress. Trends suggest that stressed systems are increasingly moving to low diversity, low mean trophic level, and a size distribution skewed to small body size (*Pauly et al., 1998*) as observed in Bocas del Toro. To maintain reef fish communities, resource managers should take factors such as sponge cover and proximity to other habitats, including mangroves, into consideration to prioritize protection efforts. Our results suggest that reef sponges and mangroves together can maintain physical structure, act as nurseries, and provide alternative habitats and thereby compensate for particular functional losses during coral mortality events. Much more information is nevertheless needed on the role of habitat connectivity if fisheries management is to be optimized and diversity hotspots safeguarded through effective spatial management that includes marine protected areas (*Linton & Warner, 2003*; *Unsworth et al., 2008*).

## ACKNOWLEDGEMENTS

We thank the divers who helped with the fish surveys and fish identification or benthic surveys, especially Scott Jones, Zachary Foltz, Ross Whippo, Justin Campbell, Jan Vincente and Seamus Harrison. We thank the people from the Bocas Research Station team for logistical help and for assistance with all aspects of the work, particularly Plinio Gondola. This is contribution 24 from the Smithsonian Institutions's MarineGEO network.

### Funding

This research was supported by the Smithsonian Tropical Research Institute, the Marine Global Earth Observatory (MarineGEO) and the Smithsonian's Tennenbaum Marine

Observatories Network, and a Nationality Council Scholarship from the University of Pittsburgh. The funders had no role in study design, data collection and analysis, decision to publish, or preparation of the manuscript.

## Grant Disclosures

The following grant information was disclosed by the authors:
Smithsonian Tropical Research Institute.
Marine Global Earth Observatory (MarineGEO).
Smithsonian's Tennenbaum Marine Observatories Network.
Nationality Council Scholarship from the University of Pittsburgh.

## Competing Interests

The authors declare there are no competing interests.

## Author Contributions

- Janina Seemann conceived and designed the experiments, performed the experiments, analyzed the data, contributed reagents/materials/analysis tools, prepared figures and/or tables, authored or reviewed drafts of the paper, approved the final draft.
- Alexandra Yingst conceived and designed the experiments, performed the experiments, contributed reagents/materials/analysis tools, authored or reviewed drafts of the paper, approved the final draft.
- Rick D. Stuart-Smith performed the experiments, analyzed the data, contributed reagents/materials/analysis tools, authored or reviewed drafts of the paper, approved the final draft.
- Graham J. Edgar performed the experiments, contributed reagents/materials/analysis tools, authored or reviewed drafts of the paper, approved the final draft.
- Andrew H. Altieri conceived and designed the experiments, contributed reagents/-materials/analysis tools, authored or reviewed drafts of the paper, approved the final draft.

## Animal Ethics

The following information was supplied relating to ethical approvals (i.e., approving body and any reference numbers):

The Ministry of the Environment Panama (MiAmbiente) and Autoridad de los Recursos Acuáticos de Panamá (ARAP) provided full approval for this purely observational research.

## Data Availability

The raw data have been provided in the Supplemental Files.

## Supplemental Information

Supplemental information for this article can be found online at http://dx.doi.org/10.7717/peerj.4455#supplemental-information.

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
