# Peer review of "The importance of sponges and mangroves in supporting fish communities on degraded coral reefs in Caribbean Panama"

_PeerJ, doi:10.7717/peerj.4455_

## Round 0.1 · original submission · Major Revisions

Both the peer reviewers are experts in this area, and both report important problems with your analytical approach and its interpretation. The reviews are substantial, so should you choose to resubmit, please be sure to provide a point-by-point response to each and every comment of both reviewers, indicating where specifically on the revised manuscript you address their concerns. I will ask the same referees to review your revision.

·

Basic reporting

The article meets the standards listed for this section (Basic Reporting) in that is is mostly clearly written. My concerns with the article are more general, and have to do with the overall approach to the topic. See notes below under General Comments.

Experimental design

I disagree with the analytical approach, and thus the nature of the research question. Since my concerns deal with the structure of the authors' approach, their experimental design, as well as their conclusions, I summarize my concerns under General Comments.

Validity of the findings

Since I don't agree with the overall approach of the authors, I don't feel than their findings are valid in the broader context that I think they need to address. Within the smaller scale context of their approach, their findings are valid.

Additional comments

I think that the authors have the makings of a very good manuscript. They have sufficient data from a broad geographic range. However, I do not think that the manuscript in current form is acceptable. My concerns are with the overall structure of their arguments, as outlined below.
General Comments:
I do not believe that general diversity indices are good proxies for estimating ecological function. For example, one can measures the same diversity for a system that is comprised of entirely different species. This can affect competition, predation, etc. The authors address this to some extent by calculating their community metrics to account for trophic level as well as disproportionately abundant species. Yet, they don’t account for the likelihood that their smaller size classes are mixtures of small-bodied fishes (which are likely residents) and juveniles (which are likely temporary). Each of these groups has different functions, will be affected differently by habitat degradation or change, not to mention the overall connectivity of the habitat mosaic. To the latter point, I suggest that small-bodied resident fishes are less likely to be affected by fragmentation (a good measure of habitat degradation) than species that ontogenetically shift among habitats. See as an example: Dahlgren, C., G.T. Kellison, A.J. Adams, B.M. Gillanders, M.S. Kendall, C.A. Layman, J.A. Ley, I. Nagelkerken, J.E. Serafy. 2006. Marine nurseries and effective juvenile habitats: concepts and applications. Marine Ecology Progress Series. 312:291-295.
A relatively diverse species assemblage that depend on different habitats than corals doesn’t necessarily translate to ecological resilience. In fact, I suggest it is more likely a phase shift, especially since different invertebrates are providing structure. If so, the discussion should be more about whether the phase shift is to a more or less complex species assemblage as well as whether it includes a trophic (or other) shift. I suggest the authors read White and Jentsch, 2001. The Search for Generality in Studies of Disturbance and Ecosystem Dynamics. Progress in Botany, Vol. 62. This article provides a good structure for examining disturbance (which is, in essence, what the authors are doing).
On a similar note, the White and Jentsch article should be helpful framing the anthropogenic disturbances that the authors are studying. In short anthropogenic disturbances differ from natural disturbances in the disturbances themselves as well as the ecological impacts from these disturbances. For example, “Does this southwestern Caribbean fish community show signals of reef degradation and over-fishing?” mixes different types of anthropogenic disturbances (direct, indirect, long-term (chronic), pulse) that one might expect to have different impacts.
How do these disturbances differ from natural disturbances (e.g., hurricanes)? Are fish communities more or less resilient or resistant to these vs natural disturbances? Would provide some overall ecological context. Might also speak to the ability of these ‘new’ systems to respond to natural disturbances.
Specific Comments:
Line 87 – “killed up to 95% of the hard coral cover” Where? In Panama, at the study site, worldwide?
Line 89 – “The persistence of invertebrate communities on these degraded reefs suggests that some resilience mechanisms are operating”. Define “resilience”. Differentiate resilience, resistance, phase shift, as in White and Jentsch 2001. I suggest that the authors consider this a phase shift since it is different invertebrate species that are now the main structure providing organisms, and that the overall fish assemblage differs from what one might call “natural”. In addition, given these differences in invertebrate communities and fish species assemblages, I would expect that this new phase will have a different level of resistance or resilience to continued anthropogenic disturbances and natural disturbances (hurricanes).
Methods
Some of my comments below may be in error due to the fact that it was unclear to me the extent that reefs, seagrass, and mangrove habitats were visually surveyed at all of the study sites (BDT as well as the Caribbean sites that are listed). Were only reef surveys conducted at the Caribbean sites, or also seagrass and mangroves?
Line 124 – were the site types paired by location (e.g., multiple Florida sites, multiple Southern Caribbean sites) or pooled across all surveys? In other words, did the authors control for potential location bias whereby differences may have been partly due to location-specific ecology? This appears more apparent in Figure 2, but is still never explained in Methods, and not really broken down in Results.
Line 130 – why were seagrass and mangroves only surveyed at BDT? Why not at other locations? How do we know whether the findings at BDT are different/same compared to other locations with or without protections (e.g., NTZ)?
Line 141 – “abundance within size classes (10 cm size bin and below; 12.5-20 cm size bins; 25 cm size bin and above)”. This makes it difficult to differentiate small bodied species from species that use seagrass/mangroves as juveniles only (e.g., Haemulidae, some Scaridae, Lutjanidae, Serranidae). Thus, it’s difficult to discern whether the non-reef habitats continue to function as juvenile habitats (even though adult habitats are degraded on the reef to the extent that juveniles may not successfully transition to adults, or at least not survive) or if these communities are mostly small-bodied resident species. This is important when considering whether this is the same community that is resilient, or if a phase shift has occurred.
Line 235 – 243 – To what extent are these results due to differences in species composition of mangrove habitats rather than anything to do with reef habitat degradation and overfishing? Perhaps surveys at other reef sites should have also included mangrove-associated habitats. Cite the article showing relationship of mangrove proximity to coral reefs and fish species composition.
Similarly, as depicted in Figure 2 – Shouldn’t the other Caribbean sites (non-BDT sites) also be examined by distance from mangroves? Since this was such a significant factor in this study, and the error bars are so large in Figure 2, if different habitats were sampled this may bias results. In addition, as the authors note (e.g., Mumby et al 2004), distance from mangroves to reef can influence species composition on the reef (due largely to the nursery value of mangroves to some species).
Line 247 – re San Andres Archipelago – was there a single site here, or multiple sites? Were similar surveys conducted in other ecoregions?
Lines 275-282 (which appear to link species composition to lack of high relief coral reef habitat) and Lines 299-303 (which now blame species composition on overfishing). Coral habitat degradation and overfishing can both causes changes in species composition and fish abundance, but the effects of each type of disturbance can differ, or if they co-occur can act synergistically. Again, see White and Jentsch for a general discussion.
Line 305-308 – what are possible causes and effects of low herbivorous species abundance? The Florida Keys reef tract has high Scaridae and Acanthuridae abundance, yet has low live coral cover and high algae cover on reefs.
Line 319-329 – this should be the sole focus of the paper. They observed different species composition due to decline in live coral cover and loss of habitat complexity. Whether to do with overfishing or other disturbances – these are potential explanatory variables there is no cause established here.
Line 334 – “Our study suggests that the positive effect of mangroves as nursery and feeding grounds can overcome and compensate some aspects of reef degradation in an ecosystem that has suffered multiple stressors.”
I’m not entirely sure of the intended meaning of this sentence. If mangroves serve as nursery grounds for species associated with the reef as adults, how does this compensate or overcome reef degradation? There is still no adult reef habitat, so ontogenetic habitat shifts can’t be completed. In other words, the juveniles that are abundant in mangrove habitat don’t represent a reproductive population that contributes to future generations if they are not able to find adult habitat which allows them to survive and reproduce (or they are harvested before they can reproduce).
Line 336 – “There are, however, non-linearities in mangrove influences on reefs, with negative influences at distances below 100 m. The reef-mangrove distance driving the highest abundance of carnivores was identified to be between 100 and 250 m.”
I am intrigued by this statement. I think that additional explanation is needed. I am not aware of studies that have documented this specific of a relationship (especially the negative influence). Mumby et al (2004) found a relationship between species composition on reefs and proximity to mangroves, but in more general terms, and not a negative relationship. Further, what is meant by “influences”, whether positive or negative? Are differences in species composition positive or negative, or merely a reflection of the habitat mosaic that supports (or doesn’t) the ontogenetic habitat requirements of different species? If the data truly show this as a result, this would, in itself, be a valuable manuscript.
Figure 2, A – are the error bars SD or SE?
Table 2 – I don’t think it’s valid to present the data at the level of species because the ontogenetic habitat requirements differ among species within the same family. For example, juvenile acanthurids of the three species observed in this study all use different habitats as juveniles (on-reef and off-reef, depending on species). So combining these species in a single factor for analysis introduces bias. See, for example:
Robertson, D. R. 1988. Abundances of surgeonfi shes on patch-reefs in Caribbean Panama: due to
settlement, or post-settlement events? Mar. Biol. 97: 495–501.
Shulman, M. J. 1985. Recruitment of coral reef fishes: effects of distribution of predators and
shelter. Ecology 66: 1056–1066.
Shulman, M.J. and J. C. Ogden. 1987. What controls tropical reef fish populations: recruitment
or benthic mortality? An example in the Caribbean reef fish Haemulon flavolineatum. Mar.
Ecol. Prog. Ser. 39: 233–242.
Shulman et al. 1983. Priority effects in the recruitment of juvenile coral reef fishes. Ecology 64: 1508–1513.
Adams, A.J. and J.P. Ebersole. 2004. Processes influencing recruitment inferred from distributions of coral reef fishes. Bulletin of Marine Science. 75(2):153-174.
Adams, A.J. and J.P. Ebersole. 2002. Use of back-reef and lagoon habitats by coral reef fishes. Marine Ecology Progress Series. 228:213-226.

Reviewer 2 ·

Basic reporting

The authors aimed to understand the factors driving the communities of fishes observed on coral reefs in Bocas del Toro and other Caribbean locations. They used a fish community assessment method designed by two of them and a large suite of environmental and biological parameters. The general approach was to look for correlations between the parameters and the fish community measurements.

I have annotated the manuscript throughout, so provide only a general synopsis below:

The manuscript is fairly well written and organized, but there quite a few places where corrections or edits are required. However, the methods and discussion fall short. Some of the comparisons that were made are difficult to assess with the given information (e.g., why would open sand in seagrass be important? This and other metrics are not mentioned in the methods) and the exact methods used to gather the information insufficient. The discussion is not terribly well organized and does not interpret some of the comparisons that were made. In some instances, broad and substantial statements are made without much discussion, or support from the results.

Experimental design

The design was not completely clear as it was not always obvious what was being measured and where. Were all metrics measured at ALL sites (reefs, seagrass, and mangroves)? Not all metrics were relevant for all sites. Many other comments are made within the manuscript.

Validity of the findings

This is really the crux of the problem with the manuscript. It is basically a study of correlations and correlations do not provide cause and effect. In too many instances the authors make assumptions about the cause of the correlations they found. In other instances, the correlations are simply correlations that do not entirely make sense and are not explained (e.g., why would proximity to mangroves drive abundance of carnivores?). See other comments within the manuscript.

Additional comments

Clearly, an enormous amount of effort went into collecting the data for this study and there is value in it. I suggest that the authors refocus their analysis on specific comparisons that makes sense and keep their interpretation to those results. I would suggest that they focus more explicitly on the fishing restriction schemes and the proximity to mangroves and seagrass beds as the parameters. The structure provided by sponges is also an interesting and compelling parameter, but describe what was measured more explicitly. If they wish to include other parameters, they should make sure they explain the reason for including them, the methods for gathering the information, and the relationship that might be expected.

Regarding interpretation, the arguments for the cause of these patterns need to be more carefully presented so they do not extend beyond the results. When speculative, this should be noted and alternative explanations should be presented.

Annotated reviews are not available for download in order to protect the identity of reviewers who chose to remain anonymous.

---

## Round 0.2 · Major Revisions

Both reviewers agree that the manuscript is much improved, yet still requires substantial revision, as noted. At this point, spelling and grammatical errors are intolerable, so please be sure that the next revision is truly publication ready. If the next revision is not fully acceptable, then I will be forced to reject this submission.

·

Basic reporting

No comment

Experimental design

The Methods are now better described, and analyses and have been adjusted to the revised approach.

Validity of the findings

The findings and conclusions are now in line with the data.

Additional comments

The authors have done a good job of addressing my comments. I think that the discussion has been stepped back to more appropriately address their results, and that the study is appropriately placed in a broader context.

There are a number of editing items that need to be addressed (misplaced words, typos). Some are addressed below, most are not. Most of my specific comments reference these editing needs. Two of my specific comments, however, I think should be addressed as part of a minor revision.

Line 13 – 14: “habitat degradation” is used as both the effect and the cause in this sentence.
Line 12: I don’t understand how the Cayman Islands can be considered part of the Bahamian ecoregion. Perhaps similar reef topography, but certainly not in the same region. And the extent of mangroves in the Turks and Caicos is much higher than the Cayman Islands. What is the justification for this?
Line 190: ‘respectively’ is misplaced here
Line 208: Regarding :”worldwide” - Wouldn’t this be true for a comparison of Caribbean and Indo-Pacific reefs, the I-P being so much more diverse in general? I suggest sticking to Caribbean comparisons. Suggest removing, as the argument can be made without it.
Lines 210 – 211: What about comparison to other OZs not in BDT, in addition to the MPAs and NTZs? Isn’t this an important part of the comparison? Are the BDT sites the same as, worse than, better than the other OZs? This is shown in the figure, but needs to be referenced in the text. Why is the BDT area worse than other OZs in the Caribbean? Is this entirely due to fishing or are other factors at BDT worse than in other locations in the Caribbean.
Line 291: “resilience” delete
Line 342: Please explain how the small size of S. iserti allows it to escape environmental pressures. I can see that its small size means it is not targeted by fishing, so it is not harvested. But what other anthropogenic pressures does it escape because of its size? Perhaps this sentence should be revised to specifically state that its small size allows it to escape pressure from fishing.

Reviewer 2 ·

Basic reporting

The authors have correct many of the problems with the writing of the manuscript, but it still suffers from a high number of grammatical errors and ambiguous wording. I have fixed many of the grammatical issues in the edited version attached. Other comments are embedded.

The methods are more clearly described, but there should be at least a brief description of the RLS method 1 protocol.

The figures are fairly well constructed, especially the map of sites, but figures 4 and 5 are not even referenced until the discussion. Figure 5 is not intuitive. There must be a better way to convey this information.

The water quality sampling is still included in the methods, although it is not in the results or discussion.

Experimental design

The experimental design is much clearer in this revised version and the goal of the research is apparent. In most cases there are details on how the data was collected an used, and where there were shortcomings.

Validity of the findings

The authors did a much better job of focusing their analysis and interpretation on relationships that made sense, instead of casting a wide net to see where it landed, as they had done prior. However, there are still questions about their interpretations and places where I think they have made assumptions or drawn conclusions not supported by the data. For example, the assumption that mangroves serve as nursery grounds for all these fish species is just an assumption. It could very well just be an alternative habitat for many. These instances are all pointed out explicitly in the edited version of the manuscript attached.

Additional comments

The manuscript is much improved, particularly with regards to the focus. However, there are still quite a few fundamental problems that need to be addressed. However, I do not think these are insurmountable.

Annotated reviews are not available for download in order to protect the identity of reviewers who chose to remain anonymous.

---

## Round 0.3 · Minor Revisions

You have done a good job responding to the reviews by the two peer referees. I have read the manuscript with a focus on clarity of presentation, attaching the annotated manuscript. There is also one conceptual revision that must be made: this is not a paper about "resilience," a topic beyond the scope of this study, so I have offered changes in wording. Please: (1) carefully incorporate the 26 (or so) minor edits I have noted in the attached annotated manuscript, (2) carefully check the manuscript for any remaining spelling and grammatical errors, and (3) carefully format the manuscript for PeerJ following the instructions to authors verbatim. I look forward to your final revisions.

---

## Round 0.4 · Minor Revisions

While most grammatical errors and misspellings were corrected, there were still some remaining, along with places where the wording was unclear. I have made as many corrections as I can in the attached manuscript file, which adds to the corrections you made previously (I will e-mail a Word version directly to Andrew Altieri, which will make preparation of the revision easy). Please accept all these changes and resubmit, making sure that everything is formatted exactly as requested by the journal.

---

## Round 0.5 · accepted · Accept

Thank you for carefully preparing your manuscript for publication. (Note: I did not write the part above about Tweeting...I hate Twitter.)